# CASS: Nvidia to AMD Transpilation with Data, Models, and Benchmark

## Abstract

We introduce CASS, the first large-scale dataset and model suite for cross-architecture GPU code transpilation, targeting both source-level (CUDA ↔ HIP) and assembly-level (Nvidia SASS ↔ AMD RDNA3 translation. The dataset comprises 70k verified code pairs across host and device, addressing a critical gap in low-level GPU code portability. Leveraging this resource, we train the CASS family of domain-specific language models, achieving 95% source translation accuracy and 37.5% assembly translation accuracy, substantially outperforming commercial baselines such as GPT-4o, Claude, and Hipify. Our generated code matches native performance in over 85% of test cases, preserving runtime and memory behavior. To support rigorous evaluation, we introduce CASS-Bench, a curated benchmark spanning 16 GPU domains with ground-truth execution. All data, models, and evaluation tools will be released as open source to foster progress in GPU compiler tooling, binary compatibility, and LLM-guided hardware translation.

## 1 Introduction

Graphics Processing Units (GPUs) are foundational to modern machine learning and scientific computing workloads due to their high-throughput parallelism. Nvidia's Compute Unified Device Architecture (CUDA) (Harris, 2024) has become the dominant programming model for GPU acceleration, but its tight coupling to proprietary hardware introduces severe vendor lock-in: CUDA code cannot run on non-Nvidia GPUs due to incompatible instruction set architectures (ISAs) (NVIDIA Corporation, 2021). As a result, organizations with large CUDA-based codebases face steep engineering costs when migrating to alternative platforms. Meanwhile, AMD GPUs, offering potential favorable performance-per-dollar (AMD, 2024a; Verge, 2024), are increasingly adopted across both data centers and consumer devices (Financial Times, 2024), creating a growing need to execute legacy CUDA programs on AMD hardware without full rewrites in software (Janik, 2024).

In response, AMD introduced the Heterogeneous-computing Interface for Portability (HIP) (AMD, 2024b), a C++ GPU API built into the Radeon Open Compute platforM (ROCm) stack (Advanced Micro Devices (AMD), 2024), designed to mirror CUDA's functionality while supporting cross-platform development. HIP enables a unified codebase for both Nvidia and AMD GPUs. Tools like HIPIFY (Advanced Micro Devices, Inc., 2025), a static translator, assist migration by converting CUDA-specific constructs into their HIP equivalents, streamlining adoption of the ROCm stack. However, HIPIFY only operates at the source level and cannot execute precompiled CUDA binaries. Furthermore, it exhibits a high failure rate when converting CUDA programs, highlighting the need for more reliable and lower-level transpilation approaches (Zahid et al., 2024).

Translating GPU assembly across vendors is hindered by divergent ISAs and compilation pipelines. Nvidia employs a proprietary toolchain centered on `nvcc`, producing PTX and low-level SASS (Harris, 2024), while AMD uses GCN/RDNA architectures compiled via the open-source ROCm stack using `hipcc` (Advanced Micro Devices (AMD), 2024) (Figure 2 provides a detailed breakdown of the alternative stacks). Bridging this gap at the assembly level is critical for democratizing the hardware computing landscape, transfer of hardware-specific optimizations across vendors, and enabling automation beyond source-level rewrites, especially for legacy CUDA codebases rich in low-level tuning. Our work introduces the first foundation for Nvidia-to-AMD assembly and source translation,

focusing on correctness and alignment. While not optimization-aware yet, it paves the way for future systems that preserve and adapt performance-critical patterns across GPU backends.

To address the lack of cross-architecture GPU translation datasets, we introduce CASS (CUDA–AMD ASsembly and Source Mapping), a large-scale corpus of 70k semantically aligned CUDA–HIP source pairs and their corresponding host (CPU – x86 ISA) and device (GPU) assemblies for Nvidia (SASS) and AMD (RDNA3) platforms. Each sample comprises functionally equivalent low-level code across vendors, verified through successful compilation and execution, enabling instruction-level analysis across execution boundaries. Unlike generic code corpora like The Stack (Lozhkov et al., 2024a), which lack GPU-aligned and compilable content, CASS provides complete source and binary representations across both GPU compute stacks. To construct CASS, we developed a fully open-source pipeline that scrapes, synthesizes, translates (via HIPIFY (Advanced Micro Devices, Inc., 2025)), compiles, and aligns GPU code. We evaluate CASS along two dimensions: (1) instruction coverage, capturing diverse SASS and RDNA3 opcodes; and (2) domain coverage, spanning real-world compute kernels from ML, graphics, and HPC. CASS is the first dataset to enable source- and assembly-level translation research for GPU architectures.

To validate the utility of our dataset, we introduce CASS-Bench, the first benchmark tailored to cross-architecture GPU transpilation. It spans 16 diverse GPU domains with execution-verified source and assembly pairs, providing a standardized testbed for future work in low-level translation and performance-aware code generation. Building on this benchmark, we present the CASS model family, a suite of domain-specific large language models fine-tuned for both source- and assembly-level GPU code translation. These models are trained on our curated corpus and demonstrate significant improvements over SoTA proprietary systems such as GPT-4o (Hurst et al., 2024), Claude-3.7 (Anthropic, 2025), and traditional tools like HIPIFY (Advanced Micro Devices, Inc., 2025)—achieving 95% accuracy in source-level translation and 37.5% in assembly translation.

Our contributions are summarized as follows:

- **CASS Dataset.** We introduce CASS, the first large-scale dataset for GPU transpilation, containing 70k semantically aligned Nvidia $\leftrightarrow$ AMD pairs at both the source (CUDA $\leftrightarrow$ HIP) and assembly levels (SASS $\leftrightarrow$ RDNA3), covering 16 real-world GPU domains.

- **CASS-Bench.** We contribute the first evaluation benchmark for cross-architecture GPU translation, with 40 curated tasks across 16 domains, including functionally verified outputs and aligned CUDA/HIP source and SASS/RDNA3 assembly.

- **CASS Models.** We release domain-specialized CASS LLMs trained for cross-architecture code translation. Our 7B model achieves 95% source and 37.5% assembly accuracy, outperforming GPT-4o and Claude (0%) on CASS-Bench. Crucially, 85% of translated assemblies preserve execution runtime and memory compared to native, confirming semantic and performance fidelity.

- **CASS Dataset Pipeline.** We designed a scalable pipeline for scraping, synthesizing, transpiling, and compiling CUDA/HIP code into aligned host and device assemblies across Nvidia and AMD GPUs.

The rest of the paper is organized as follows: §2 reviews prior work on Nvidia-to-AMD and assembly translation. §3 describes our data collection, conversion, and filtering pipeline. §4 analyzes dataset structure and coverage. §5 outlines model training and evaluation, with results and ablations in §6. Finally, §7 lists limitations and future work, followed by §8 concluding remarks.

## 2 RELATED WORKS

In this section, we describe prior work in GPU translation efforts (§2.1), assembly-level transpilation (§2.2), and related benchmarks (and their shortcomings) in the space (§2.3).

### 2.1 TRANSLATING FROM NVIDIA TO AMD

The fragmentation of GPU software ecosystems has driven the need for robust CUDA-to-HIP translation tools. HIPIFY (AMD ROCm Documentation, 2024) statically converts CUDA source code into HIP source code, enabling ROCm compatibility via direct syntax substitution. Operating at a lower abstraction, CuPBoP-AMD (Chen et al., 2023) translates NVVM IR to HIP-compatible LLVM IR

Table 1: Comparison of Domain/Characteristics across Different Datasets

| Domain/ Characteristics | ComputeEval NVIDIA | Rodinia Bench | SHOC | Poly Bench | Babel Stream | Ours |
|---|:---:|:---:|:---:|:---:|:---:|:---:|
| CUDA (source) | ✓ | ✓ | ✓ | ✓ | ✓ | ✓ |
| OpenCL (source) | ✗ | ✓ | ✓ | ✓ | ✓ | ✓ |
| SASS (assembly) | ✗ | ✗ | ✗ | ✗ | ✗ | ✓ |
| RDNA3 (assembly) | ✗ | ✗ | ✗ | ✗ | ✗ | ✓ |

using the LLVM toolchain (Lattner & Adve, 2004; The Clang Team, 2025), offering more flexible intermediate-level interoperability. Earlier, GPU Ocelot (Diamos et al., 2009) explored dynamic binary translation, recompiling CUDA to AMD/x86 ISAs at runtime. Although innovative, it was limited by poor scalability and high overhead, making it impractical for modern GPU workloads. All these tools have lacked consistent updates to keep up with CUDA advances, suffer from usability issues, and operate only at the source level.

More recently, ZLUDA (Janik, 2024) introduced a runtime system for executing unmodified CUDA binaries on AMD GPUs without source access by intercepting CUDA APIs and translating PTX/SASS into AMD-compatible code via LLVM. Originally targeting Intel, it now supports AMD RDNA3 through runtime patching. ZLUDA operates at the LLVM IR level rather than the hardware assembly. While a reasonable level in the stack to target, ZLUDA would not be able to benefit from low-level, backend Nvidia optimizations (operating below the PTX level), and is limited to the AMD stacks backend optimizations. In our work, we target assembly-to-assembly translation, in an effort to leverage hardware-specific optimizations below the intermediate representation (IR) level, that may be missing altogether in the corresponding AMD codebase.

## 2.2 ASSEMBLY-TO-ASSEMBLY TRANSLATION

Translating assembly across ISAs is challenging due to divergent instruction sets and execution models. Recent work employs language models for this task, including CRT (Heakl et al., 2025), a lightweight transpiler from x86 assembly (CISC) to ARM (RISC), and Guess & Sketch (Lee et al., 2023), which integrates language models with symbolic reasoning to translate between ARMv8 and RISC-V. These recent successes open the door for assembly-to-assembly translation in the unexplored GPU-to-GPU space. A key contributing factor to their success is the large CPU-centric dataset enabling training from one ISA to another. Given the lack of such a rich dataset in the GPU space, a primary goal of this work is to enable such an exploration and transpilation across GPU vendors, democratizing compute in the critical GPU and ML-acceleration landscape, where Nvidia/CUDA currently dominate the market.

## 2.3 DATASETS AND BENCHMARKS FOR CUDA AND HIP

As shown in Table 1, existing benchmarks in the GPU space generally focus on runtime performance, do none target the assembly level, and do not have paired/aligned data across Nvidia/AMD codebases. ComputeEval (NVIDIA, 2024) includes only CUDA code for hardware evaluation. Rodinia (Che et al., 2009) and SHOC (Danalis et al., 2010) provide heterogeneous benchmarks using CUDA/OpenCL/OpenMP but omit AMD code and assembly. PolyBench (Grauer-Gray et al., 2012) evaluates compilers with CUDA/OpenCL kernels, yet lacks assembly-level or AMD support. BabelStream (Deakin et al., 2016) benchmarks HIP/CUDA/OpenCL memory bandwidth but excludes assembly and domain diversity. Hetero-Mark (Sun et al., 2016) targets CPU–GPU workloads where GPU code is minimal. The Stack (Lozhkov et al., 2024a;b) dataset nearly 200k CUDA files but no AMD coverage or aligned assembly. In contrast, CASS uniquely offers 70k semantically aligned CUDA–HIP source and SASS–RDNA3 assembly pairs across both host and device, enabling instruction-level analysis and forming the first dataset purpose-built for cross-vendor GPU assembly translation.

To the best of our knowledge, no existing dataset provides *paired* source- and assembly-level Nvidia-AMD code, hindering effective training and benchmarking.

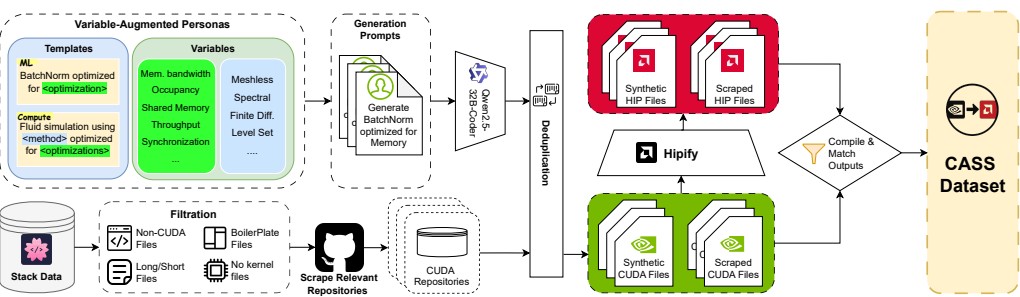

Figure 1: `CASS` pipeline: We collect CUDA code from public repositories and synthesize additional samples via prompt-based LLM generation. After filtering and deduplication, all CUDA files are translated to HIP using HIPIFY, then compiled to extract host and device assembly. Matched outputs form the `CASS` dataset with aligned source and assembly pairs across Nvidia and AMD stacks.

## 3 METHODS

This section outlines the end-to-end methodology behind `CASS`, including data collection, code conversion, and compilation for Nvidia and AMD GPUs. We built the low-level assembly corpus from high-level CUDA code using three strategies: scraping public repositories, generating synthetic samples, and applying targeted code generation frameworks.

### 3.1 CUDA CODE SCRAPING

We leveraged the Stackv2 dataset (Lozhkov et al., 2024b) to extract CUDA source files. This dataset, curated from a vast array of public code repositories, offers deduplicated and license-compliant samples, facilitating the assembly of a diverse corpus of GPU-oriented code. To maximize the number of compiled files in the later stage, we used the dataset's metadata to identify and download the top 200 repositories with the largest number of CUDA files. This repository-level download preserved the original directory structure and relative imports, as shown in Figure 1, and improved compilation success by 23.7% compared to isolated file scraping. After extraction, we applied additional filtering to remove overly long files ($> 7$k lines), trivially short files ($<10$ lines), naive boilerplate samples (e.g., "Hello World"), and files lacking CUDA kernel definitions. This process resulted in a final set of 24k usable samples.

### 3.2 SYNTHETIC DATA GENERATION

To circumvent the issue of low architectural and semantic diversity in underrepresented GPU kernels from "real-world" code pairs, we employed a coding-oriented large language model (`Qwen2.5-Coder32B`) to synthesize CUDA kernel code using our variable-augmented persona strategy. We found this important because the amount of GPU code online in general is limited, both in quantity and diversity, and hence one of the goals of `CASS` is to address this problem for ourselves and others in the space.

The process begins by defining a set of natural language prompt templates with variable placeholders. For example, a template might read:

> *Generate a CUDA kernel for cloth simulation with a {size}X{size} grid. Optimize for {optimization}.*

To fill these templates, we prepared predefined lists of variable values. For instance, {`size`} was instantiated with values such as 32, 64, and 128, while {`optimization`} was sampled from options like "memory bandwidth", "register usage", and "multi-GPU scaling". This allowed us to systematically generate a broad range of prompts, each specifying different values for the placeholders in the templates. Appendix A.8 includes full details on prompts, variables, and value ranges used for synthetic data generation.

These prompts were then passed to the LLM, which generated CUDA source files accordingly. While this method introduced some functional inconsistencies that required significant post-generation filtering (syntactic errors, missing definitions, or invalid memory operations), it enabled the creation

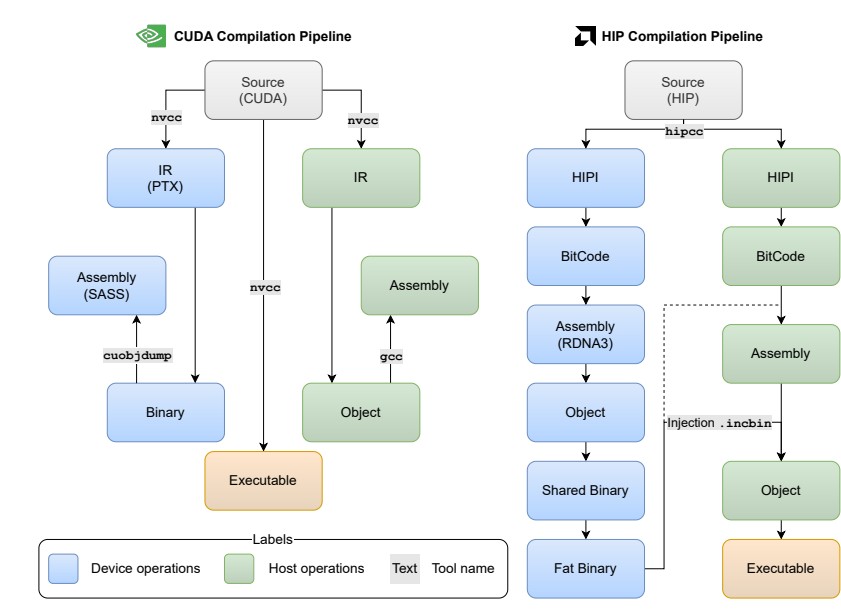

Figure 2: The Nvidia (left) and AMD (right) stacks illustrate the compilation process for CUDA and HIP. Blue denotes device-side steps; green denotes host-side steps. Nvidia's stack is opaque; accessing device assembly (SASS) requires first compiling to binary, then using cuobjdump. In contrast, AMD's process is transparent, allowing direct inspection and modification of device assembly (RDNA3) before host integration.

of rich and diverse CUDA samples. In total, we generated 85k CUDA samples, of which 49.1% compiled successfully, yielding a final set of 46.3k valid files of synthetic data (complementing the 24k "real-world" date described in §3.1).

## 3.3 TRANSPILATION AND COMPILATION

After collecting CUDA files from the previous stages, we performed deduplication to ensure all samples are unique in our dataset. We then used AMD's Hipify tool (Advanced Micro Devices, Inc., 2025) to convert CUDA source files by replacing CUDA-specific API calls with HIP equivalents. Files that failed conversion (approx. 43.9%) were discarded. Once CUDA–HIP pairs were available, we compiled them to host and device assemblies using -Os compilation flag to reduce code size, achieving a 9.3% average token reduction compared to O3. Given the architectural divergence of the two stacks (see Figure 2), their compilation pipelines differed substantially, requiring significant effort to engineer and standardize our described workflow.

In Figure 2, a key distinction between the CUDA and HIP compilation pipelines lies in how they manage host and device assembly separation. In ROCm, the device binary is typically embedded into the host binary during the BitCode-to-assembly transition. We modified this behavior by deferring insertion until after host assembly was converted to object code, enabling: (1) independent extraction of pure host (CPU) and device (GPU) assemblies, and (2) selective recombination for controlled translation and evaluation.

Conversely, Nvidia provides no access to its binary injection process, device and host assemblies remain intertwined, with no official method for extraction or reintegration (NVIDIA Corporation, 2025). Since our goal was to support host-to-host and device-to-device transpilation, recombination on the CUDA side was unnecessary. Instead, we developed a regex-based filtering pipeline to disentangle host and device assembly sections during CUDA compilation.

After compiling both stacks to SASS and RDNA3, we retained only samples that compiled successfully on both Nvidia and AMD pipelines, accounting for asymmetric failures. The final dataset includes matched CUDA–HIP source pairs, SASS–RDNA3 device assemblies, and host assemblies. In total, 64k samples were collected after this stage.

### 3.4 OpenCL Pipeline

OpenCL stands as an independent pipeline in generating Nvidia to AMD mapping datasets outside of the CUDA/HIP framework. In other words, it compiles down to the assembly level without going through the aforementioned stacks, operating as a single "source" for GPU code deveolpment (Group, 2025). Approximately 6k OpenCL code snippets were collected from the Stack dataset and compiled down to the device assemblies. On the Nvidia stack, a wrapper C++ function was used to encapsulate the clBuildProgram library provided by OpenCL (Group, 2020) and convert them into PTX, after which the CUDA stack was used to compile them down to assemblies. On the AMD stack, clang was used to directly transpile the OpenCL files into device assemblies whilst forcing it to emit intermediate LLVM during this process (The Clang Team, 2025).

Table 2: Dataset composition by source and size

| Dataset | Collected | Final |
|---|---|---|
| Synthetic | 85.5k | 40.6k |
| Stack | 124.1k | 24.1k |
| OpenCL | 6.6k | 5.9k |
| **Total** | - | **70.7k** |

The final instruction training dataset (CASS) comprises 70,694 aligned samples spanning a broad range of domains, with a primary focus on GPU compute and GPU-related data structures (Figure 4, Table 2). Each sample includes both CUDA and HIP source code alongside its compiled assembly representation, with pairwise source/assembly alignments verified to compile successfully. All compilations were performed on an Nvidia A100 PCIe machine for the CUDA stack (SASS sm85 ISA) and on AMD Radeon RX 7900 XT GPUs (RDNA3 ISA) for the AMD stack.

## 4 CASS AND CASS-BENCH DATASETS

This section presents and details our two complementary datasets: the large-scale CASS corpus and the evaluation-focused CASS-Bench. We first analyze CASS to characterize structural divergences between CUDA and HIP at both source and assembly levels, examining factors such as code length, syntactic similarity, and opcode diversity. We then introduce CASS-Bench, a curated benchmark spanning diverse GPU domains with source–assembly pairs, designed to provide a common ground for evaluating cross-architecture translation methods.

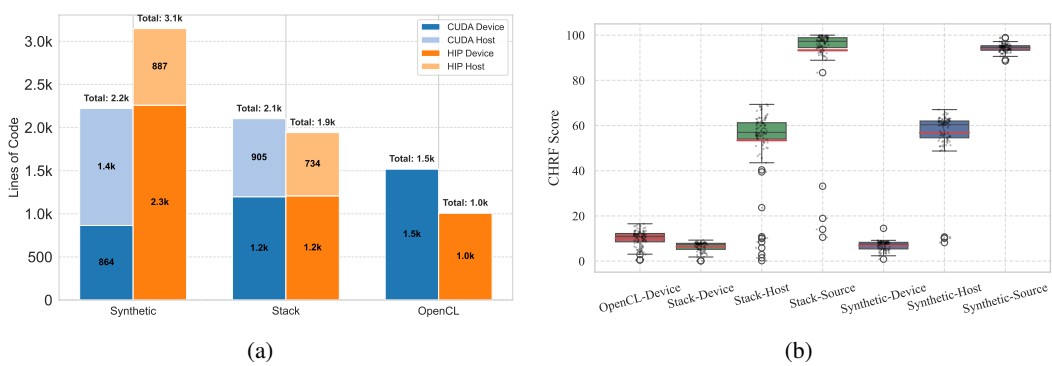

(a)          (b)

Figure 3: Comparison of structural and syntactic patterns in CASS: (a) verbosity across subsets and backends; (b) syntactic similarity of translated code.

### 4.1 Dataset Analysis

CASS reveals pronounced structural divergence between CUDA and HIP at both source and assembly levels, underscoring the inherent complexity of cross-architecture GPU transpilation. We analyze this by looking at the length of the assembly files, their syntactic similarity, and opcode diversity.

**Length of Assembly Files.** Figure 3 (left) shows that AMD device assembly is, on average, twice as long as Nvidia's in both synthetic and Stack subsets, while Nvidia's device assembly exceeds HIP device assembly by approx. 50% in the OpenCL set. We found an exponential relationship between source complexity and assembly size, with CUDA producing more verbose outputs than HIP for equivalent code. This highlights the growing difficulty of assembly-level translation as code complexity scales. See appendix A.5.1 for full details.

**Syntax Similarity**. As illustrated in Figure 3 (right), the CHRF (Popović, 2015) score, which measures character-level similarity between sequences, indicates that HIP and CUDA assembly exhibit low syntactic similarity for both device and medium similarity to host code, particularly in the OpenCL and Stackv2 subsets. In contrast, the source code translations, especially in the synthetic subset, show high overlap, highlighting that surface-level syntax is better preserved in the source code than in the compiled assembly representations.

**Opcode Diversity**. We noticed that tensor operations dominate both CUDA and HIP assembly, especially in device code, with memory-related instructions such as `mov` and `call` appearing most frequently (refer to appendix A.5). Additionally, HIP opcodes like `s_mov_b32` and `v_add_co_u32` are used extensively reflecting low-level vector and memory operations unique to AMD's ISA, while Nvidia is dominated by its own variant of common instructions such as `movq`, `call`, and `jmp`, with greater host-side integration (refer to appendix A.5). Both stacks share common control and memory ops (e.g., `mov`, `test`), but HIP provides finer-grained access to GPU internals, revealing deeper visibility into parallelism. The synthetic subset emphasizes memory-oriented instructions, aligning with LLM-driven template optimizations. We further conduct a t-SNE visualization of opcode embeddings generated by BERTCoder to examine the semantic relationship between Nvidia and AMD instructions. The resulting clusters indicate that, despite differences in backend implementations, the two vendors exhibit semantically aligned opcode distributions across both device and host levels. A more detailed view of these clusters is provided in the appendix (Figure 6).

## 4.2 CASS-BENCH

`CASS-Bench` was created to provide a standardized benchmark for assembly-to-assembly transpilation, allowing fair evaluation and comparison of models on execution-verified cross-architecture tasks. The benchmark is a 40-sample evaluation suite spanning 16 GPU-centric domains, each represented by 1–5 curated prompts. For each, we (1) used `Claude-3.7` to generate a CUDA implementation; (2) compiled and executed on Nvidia hardware to obtain reference outputs; then (3) prompted `Claude-3.7` to generate the corresponding AMD code. If outputs mismatched due to compilation errors, formatting differences or random generators variance, the AMD code was regenerated. Only samples with manually verified output equivalence were included. All final Nvidia–AMD pairs were processed using our pipeline (§3) to extract aligned host and device assembly. Figure 4 (right) shows the category distribution of `CASS-Bench`.

## 5 EXPERIMENTS

We evaluate the `CASS` dataset by instruction-supervised fine-tuning the `Qwen2.5-Coder` (Hui et al., 2024) models at various parameter scales. Two variants are developed: one for assembly translation (SASS → RDNA3) and another for source translation (CUDA → HIP). We benchmark these models against both proprietary and open-source baselines, including larger-scale systems.

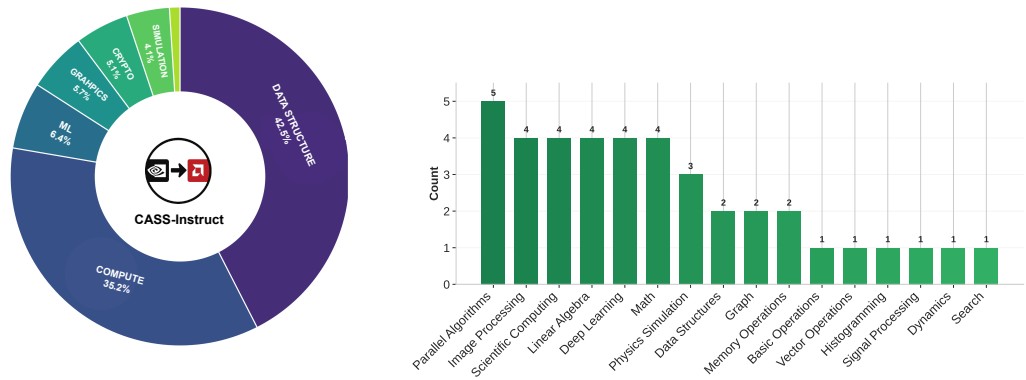

Figure 4: `CASS` coverage across dataset and benchmark (left) domain distribution of training samples (right) category distribution in `CASS-Bench`.

Table 3: Performance of different models on our CASS-Bench. Bold cells refer to the best results.

| | Model | Assembly Accuracy (%) | Source-to-Source Accuracy (%) |
|---|---|---|---|
| **Tools** | ZLUDA | 2.5% | 27.5% |
| | Hipify | – | 87.5% |
| **LLMs** | GPT-4o | 0% | 90.0% |
| | Gemini-2.0-Flash | 0% | 80.0% |
| | Claude-3.7 | 0% | 90.0% |
| | Qwen2.5-Coder-32B | 25.0% | 85.0% |
| **Ours** | CASS-1.5B | 12.5% | 90.0% |
| | CASS-3B | 20.0% | 92.5% |
| | CASS-7B | **37.5%** | **95.0%** |

**Instruction Supervised Finetuning.** To ensure that input samples fit within the 16K-token context window of the LLM, we normalized CUDA assembly code by removing redundant whitespace and comments, which reduced token count by roughly 15%. No preprocessing was applied to HIP assembly code due to its sensitivity to whitespace changes. We fine-tuned the `Qwen2.5-Coder` models at 1.5B, 3B and 7B parameter scales on 4xA100 GPUs, using a batch size of 4, gradient accumulation of 32 (effective batch size of 512) and a learning rate of $1 \times 10^{-5}$. The relatively aggressive learning rate was selected due to the dataset's distributional divergence from the models' pretraining corpus. Training employed DeepSpeed (Rasley et al., 2020) with optimizer state sharding to maximize hardware efficiency, achieving 98% GPU utilization. Additionally, we incorporated Liger Kernel (Hsu et al., 2024) and Paged Adam optimizer (Loshchilov & Hutter, 2017) to accelerate training and manage memory more effectively. We utilized LLaMA-Factory (Zheng et al., 2024) to implement all of these optimizations. All models were trained with a 16K-token context window. At inference time, we applied RoPE (Su et al., 2024) extrapolation to support up to 32.7K tokens. Inference was efficient, requiring approximately 56 seconds per a 16K-token sample.

**Evaluation Protocol.** For both source and assembly transpilation, the LLM-generated code (HIP source or host/device assembly) was compiled and executed. The resulting outputs were then compared against the ground truth from CASS-Bench to verify functional correctness.

## 6 RESULTS

**Assembly-to-Assembly Performance.** Table 3 reports `CASS-Bench` results across LLMs and tools. All baselines, including proprietary and large open models, such as GPT-4o (Hurst et al., 2024), Gemini-2.0-Flash (Hassabis & Kavukcuoglu, 2024), and Claude-3.7 (Anthropic, 2025), failed with 0% accuracy, except Qwen2.5-Coder-32B (Hui et al., 2024), which reached 25%. ZLUDA (Janik, 2024), a runtime-level system, achieved only 2.5% assembly accuracy despite operating directly on compiled binaries, which may be attributed to its compatibility with RNDA1. In contrast, our `CASS` models reached up to 37.5%, surpassing all the baselines by a large margin and highlighting that our dataset imparts essential assembly-level knowledge absent from existing tools and models.

**Code Efficiency and Analysis.** Assembly accuracy varies across domains, with 0% in math, data structures, and graph tasks, 25–50% in linear algebra and memory operations, and up to 100% in physics simulations—highlighting the challenge of preserving low-level semantics. Despite this, the translated code closely matches the original in execution: memory usage deviates by less than $\pm0.3\%$, and execution time stays within $\pm11.8\%$, with over 85% of samples falling within $\pm5.6\%$ for both metrics, confirming that our model preserves both memory and runtime efficiency during assembly translation. Each test was executed 20 times, and the reported values reflect the average across runs to mitigate noise and ensure statistical reliability. For more details, refer to Figures 5 and 7 in the appendix.

**Source-to-Source Performance.** To further validate the utility of the dataset, we also evaluated source transpilation performance as shown in Table 3. This task aligns more closely with some of the pretraining

objectives of many proprietary models, as reflected in their relatively strong performance (ranging from 80% to 90%). Nonetheless, even the smallest CASS model (1.5B) significantly outperformed all baselines, achieving 90% accuracy. The 7B variant showed an outstanding state-of-the-art performance of 95% accuracy. Although our CUDA dataset was entirely translated by Hipify and we retained only semantically aligned samples, our model surpassed Hipify's source-to-source capability by 7.5%.

**Ablation Study.** Table 4 shows that using only The Stack data yields 17.5% assembly accuracy. Adding synthetic data improves it by +12.5%, highlighting its role in learning low-level patterns. OpenCL adds +2.5%, providing complementary coverage, while RoPE extrapolation pushes accuracy to 37.5% by extending context capacity.

Table 4: Ablation study on the impact of different data.

| Experiment | Source Accuracy | Assembly Accuracy | Δ Impact |
|---|---|---|---|
| Stack subset | 87.5% | 17.5% | - |
| +Synthetic | 95.0% | 30.0% | +12.5% |
| +OpenCL | 95.0% | 32.5% | +2.5% |
| +RoPE Extrapolation | 95.0% | 37.5% | +5.0% |

**Hardware Generalization.** To demonstrate the generalizability of our methodology across GPU architectures, we extend our pipeline to a second device pairing (RTX4090 ↔ RX7900 XT). Results show similar generalization as our first GPU pair, with 32.5% assembly accuracy, confirming the feasibility and challenges of ISA variations. Additionally, it showcases the need for a `CASS` and `CASS-Bench`, as data is a limiting factor in both the training and the inference stages. Nevertheless, we envision that our proposed methodology and code will enable future contributions for more crowd-sourced ISA pairings.

## 7    LIMITATIONS AND FUTURE WORK

While our work establishes a strong foundation for cross-architecture GPU translation, several points of improvement remain. The current assembly translation accuracy, though state-of-the-art, highlights the inherent difficulty of low-level code generation, a challenge we explicitly address with the introduction of `CASS-Bench`, the first benchmark for this task. We hope it will catalyze future research into more robust translation models.

Furthermore, while the `CASS` dataset is the largest of its kind, covering 70k aligned samples, real-world deployment may benefit from even broader architectural and domain coverage. Our dataset provides a critical starting point for data-efficient training in a field where high-quality, aligned examples are scarce.

Finally, the 16K-token context window limited the inclusion of certain vendor-specific optimizations. We see this as an opportunity for future work to explore longer-context models or hierarchical compilation strategies, building on the pipeline and alignment methodology introduced here.

## 8    CONCLUSION

We present CASS, the first large-scale dataset and model suite for cross-architecture GPU code transpilation, encompassing 70k aligned pairs of source and assembly code for both Nvidia and AMD platforms. Our dataset uniquely bridges both source-to-source (CUDA to HIP) and assembly-to-assembly (SASS to RDNA3) mappings, addressing a critical gap in low-level code portability. To validate its effectiveness, we train the CASS model family, which achieves 95% accuracy in source translation and 37.5% in assembly translation, substantially outperforming both proprietary and open-source baselines. Furthermore, our transpiled code preserves functional behavior: over 85% of samples match native execution in both memory usage and runtime. We also introduce CASS-Bench, a purpose-built evaluation suite spanning 16 GPU-centric domains. All models, data, and benchmarks are released as open-source resources, establishing a foundation for future research in compiler tooling, hardware interoperability, and performance-aware code generation.

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

# A APPENDIX

## A.1 COMPUTATIONAL RESOURCES AND ENVIRONMENTAL IMPACT

### A.1.1 HARDWARE USED

All experiments were conducted on two distinct machines to generate architecture-specific outputs. For AMD-related compilation and execution, we used a workstation equipped with an Intel i7-14700KF CPU and an AMD Radeon RX 7900 XT GPU. For Nvidia-related outputs, we used a server with an AMD EPYC 9654 CPU and an Nvidia A100 (80GB) GPU. Furthermore, to ensure consistency and reproducibility across platforms, all file generation was performed within Docker containers tailored to each architecture.

### A.1.2 CARBON FOOTPRINT

Energy consumption was measured using the `CodeCarbon` tool during fine-tuning experiments conducted on $4 \times$ A100 (40GB) GPUs for 2 epochs across 70,000 samples. The results are as follows:

- **Qwen2.5-Coder 1.5B.** 20.13 kWh, corresponding to approximately 12.11 kg $CO_2$ emissions;
- **Qwen2.5-Coder 3B.** 34.27 kWh, corresponding to approximately 20.72 kg $CO_2$ emissions.

These measurements align with expected energy usage benchmarks for models of similar size and training duration.

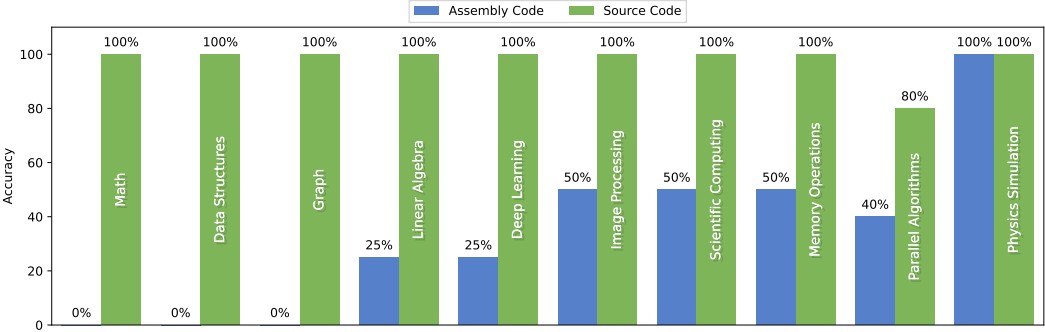

Figure 5: Source and assembly-level accuracy across categories.

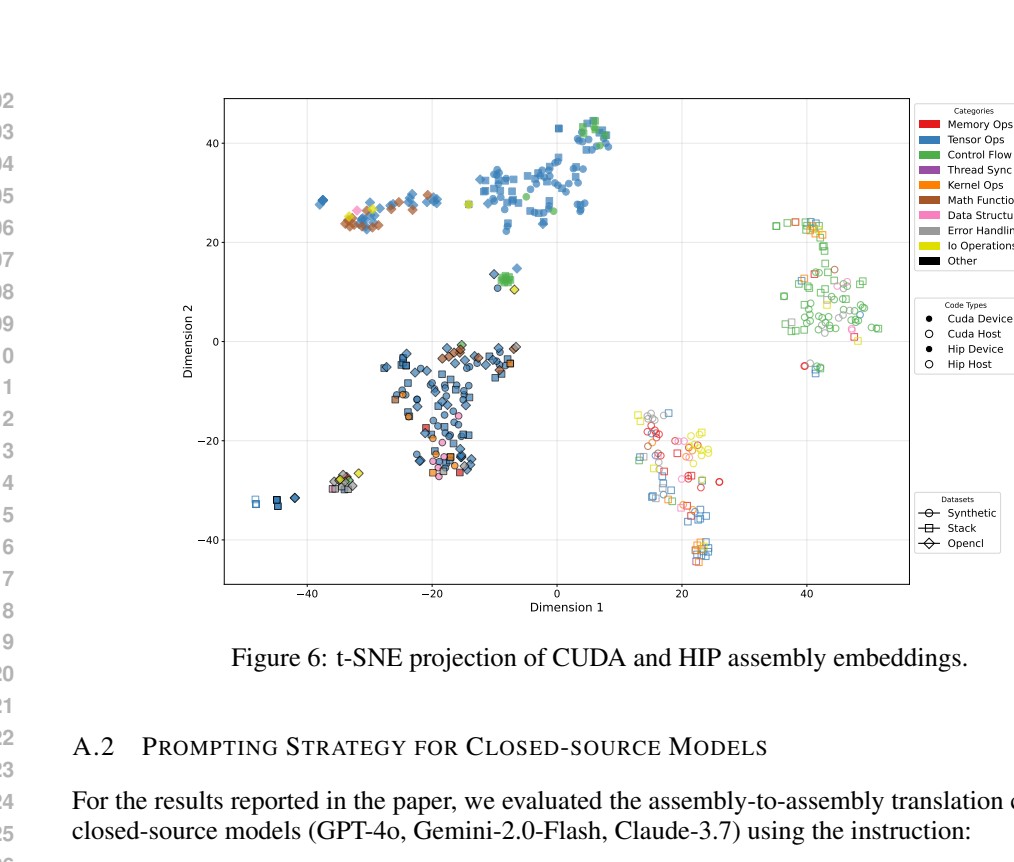

Figure 6: t-SNE projection of CUDA and HIP assembly embeddings.

## A.2 PROMPTING STRATEGY FOR CLOSED-SOURCE MODELS

For the results reported in the paper, we evaluated the assembly-to-assembly translation capacity of closed-source models (GPT-4o, Gemini-2.0-Flash, Claude-3.7) using the instruction:

> *You are given a CUDA assembly (SASS) code and you are required to convert it into HIP assembly (RDNA3) code without changing the functionality. The code output from CUDA and HIP should be the same when executed.*

We also experimented with more advanced prompting strategies, adding few-shot examples of (SASS, RDNA3) pairs and applying chain-of-thought (CoT) prompting, but observed no significant performance gains. This outcome likely stems from the limited prior exposure these models have low-level GPU assembly code during pretraining. Even with better prompting, the models lack the internal structure or inductive bias needed to reason over hardware-specific instruction patterns.

## A.3 EVALUATION ON ZLUDA

To assess ZLUDA's ability to execute CUDA code on AMD GPUs, we designed a two-track evaluation strategy targeting both source-level and binary-level workflows (the latter being akin to assembly-level translation). In the source-to-source setting, we leveraged access to the original CUDA source files to manually compile them into PTX using `nvcc`. These PTX files were then ingested by ZLUDA, which translated them into AMD-compatible LLVM IR before lowering them into native executables targeting RDNA3 hardware. In the assembly-to-assembly setting, we instead compiled the CUDA source into a complete executable and invoked it directly. ZLUDA intercepted the CUDA runtime calls, dynamically translated the embedded PTX or SASS, and executed the resulting code on the AMD backend. This dual strategy allowed us to assess both ZLUDA's static translation capabilities and its runtime interoperability under realistic execution conditions.

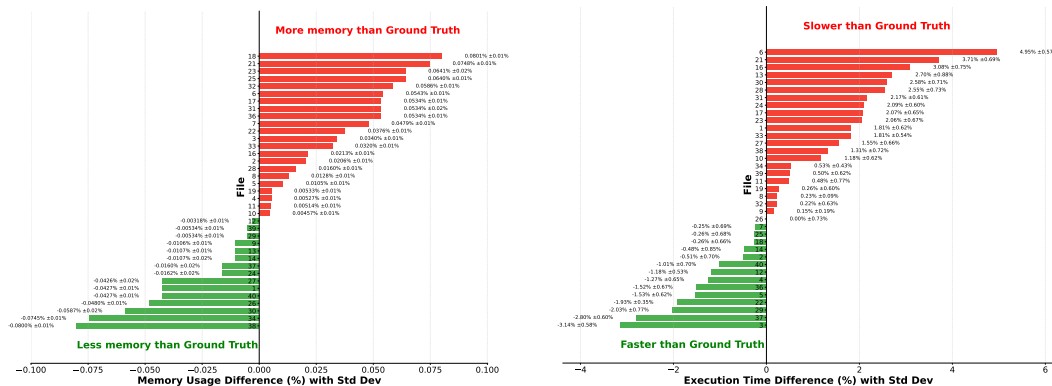

Figure 7: Comparison of memory usage (left) and execution time (right) between predicted and ground truth HIP programs, measured via compilation and runtime profiling.

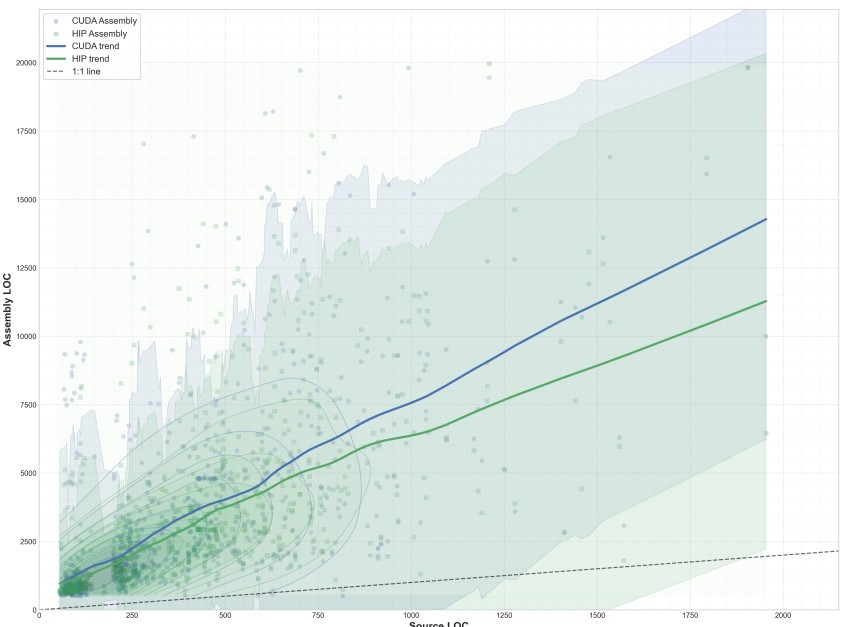

Figure 8: Relationship between source and assembly-level LoC in the CASS dataset. Scatter plot comparing source code lines of code (LoC) to the corresponding assembly LoC for both CUDA and HIP backends across the Stackv2 and Synthetic subsets. Trend lines and density contours illustrate that CUDA typically produces more verbose assembly output than HIP for equivalent source sizes.

## A.4 CASS DOMAIN COVERAGE

To obtain the domain-level breakdown shown in Figure 4, we developed a static analysis pipeline that categorizes each source file based on its content. The classification is performed by matching the file's text against curated sets of domain-specific keywords corresponding to seven high-level categories: *general compute, simulation, data structure, machine learning, graphics, cryptography,* and *scientific computing*. Each keyword set includes terms commonly associated with the respective domain; for example, the *machine learning* category includes terms such as neural, gradient, and activation, while *cryptography* includes hash, encrypt, and signature. For a given file, the domain with the highest keyword match count is assigned. If no keywords are matched, a default label (e.g., *general compute*) is applied. After all files are processed, their assignments are aggregated to produce the final domain distribution. This process provides a simple yet straightforward and interpretable way of grouping source files by their functional domain.

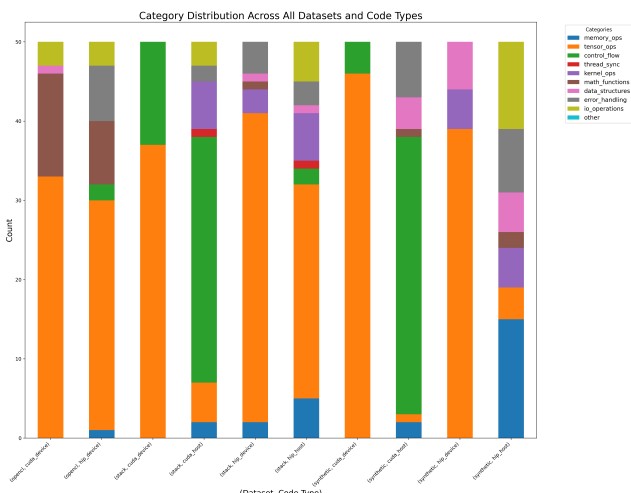

Figure 9: Opcode Category Distribution by Dataset and Code Type. Stacked bar chart showing the distribution of assembly instructions across 10 opcode categories for device and host code in the Synthetic, Stackv2, and OpenCL subsets. Each bar represents a (dataset, code type) pair, illustrating the functional composition of the code across memory, tensor, control flow, synchronization, and other operations.

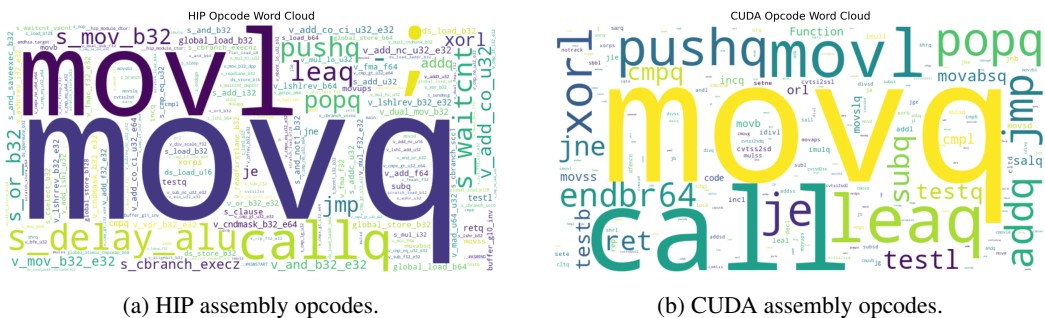

(a) HIP assembly opcodes.                    (b) CUDA assembly opcodes.

Figure 10: Most frequent opcodes in HIP and CUDA assembly. Word clouds depicting the most common opcodes in HIP and CUDA assembly files. The size of each opcode reflects its relative frequency in the compiled dataset, highlighting structural and architectural differences between the two backends.

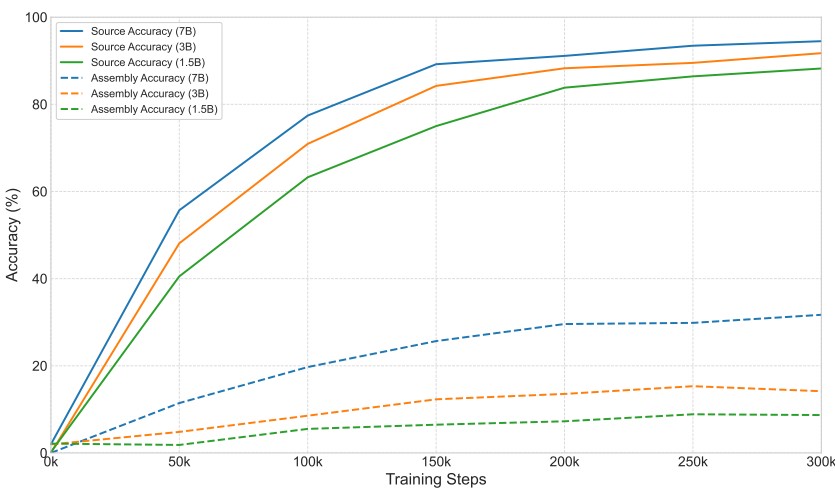

Figure 11: Accuracy vs. training steps for source/assembly across CASS model scales (1.5B, 3B, 7B).

## A.5 Extra Data Analysis

### A.5.1 Length of Assembly Files

As shown in the Figure 8 We found an exponential relationship between source complexity and assembly size, with CUDA producing more verbose outputs than HIP for equivalent code. This highlights the growing difficulty of assembly-level translation as code complexity scales.

### A.5.2 Opcode Diversity

Taking a deeper dive into the low-level instructions representation shown in Figure 10, a few extra insights can be drawn. In the HIP case, many opcodes, such as `s_mov_b32`, `v_add_co_u32`, and `s_waitcnt`, come directly from AMD's GPU instruction set. These reflect fine-grained control over the hardware, including scalar and vector operations and synchronization. On the other hand, the CUDA assembly is mostly made up of x86-64 instructions like `movq`, `call`, `jmp`, and `pushq`, which are typically used on the CPU. This suggests that the CUDA output includes more host-side code or that GPU instructions are hidden behind a higher level of abstraction. Still, both stacks share common instructions like `mov` and `test`, showing that some basic control and memory operations are similar. In general, HIP provides more visibility into what the GPU is doing, while CUDA hides many of those low-level details behind a more unified host-device model.

## A.6 Performance Degradation Analysis

We analyzed failed kernel translations and categorized them by key operation types. Among the failed cases, we found: 100% involved control flow (e.g., `for`, `while`), 75% accessed global memory, 62.07% used synchronization (e.g., `__syncthreads()`), 10.34% involved atomic operations (e.g., `atomicAdd`), 6.82% used shared memory, and 11.36% included local arrays. It's important to emphasize that these categories are not mutually exclusive, as most failed files involve multiple overlapping operation types. Moreover, as shown in Figure 5 in the paper, assembly-level failures are concentrated in Math, Data Structures, Graph, and parts of ML domains, indicating that control-heavy or abstract computation tasks remain the most challenging for the model. These trends suggest that failures are strongly correlated with increased kernel complexity, particularly in terms of global memory access and memory synchronization, which likely strain the model's limited context and structural understanding at the assembly level.

## A.7 Stratified Benchmarking

We performed a stratified evaluation of the assembly benchmark by grouping samples based on input length measured in tokens: easy (<9k), medium (9k-12k), and hard (>12k). The model's accuracy decreases with input length, 35.0% (easy), 33.33% (medium), and 17.65% (hard), highlighting that longer sequences pose greater challenges, likely due to context compression.

## A.8 Sythetic Generation

To generate large-scale, diverse CUDA programs, we design a multiprocessing Python pipeline that interacts with a locally hosted large language model via a chat-based API. The pipeline leverages a wide array of handcrafted prompt templates, each parameterized with variables such as problem size, optimization target, algorithm type, and architectural features (see Appendix A.8.1). At runtime, these templates are instantiated with randomly sampled values from curated sets covering domains like matrix operations, graph algorithms, scientific computing, machine learning, and sparse computation (see Table 5). Each worker process independently generated prompts, sends them to the model, extracts valid CUDA code from the response, and saves the output in a structured format. Robust fault-tolerance mechanisms—including retry logic, output validation, and file existence checks—ensure resilience to model failures and concurrent access.

Additionally, to avoid reproducing training data seen by the LLM, we apply both prompt-space and output-space deduplication: (1) prompt templates are checked for novelty against LLM training corpora, where verifiable, and (2) generated samples are structurally parsed and filtered using AST and opcode similarity to eliminate near-duplicates. The system supports parallel generation with controlled

API concurrency and automatic resumption from previous checkpoints, enabling scalable and efficient generation of compilable CUDA code samples suitable for downstream benchmarking or training.

Table 5: Representative values for prompt placeholders used in the synthetic code generation.

| Placeholder | Example Values |
|---|---|
| {size} | 64, 1024, 16384 |
| {dimension} | 1, 3, 6 |
| {optimization} | memory coalescing, shared memory usage, warp-level programming |
| {operation} | sum, histogram, L2 norm |
| {algorithm} | matrix multiplication, radix sort, BFS |
| {radius} | 1, 5, 13 |
| {graph_format} | adjacency matrix, CSR, edge list |
| {md_algorithm} | Verlet integration, leapfrog, Runge-Kutta |
| {linear_solver} | conjugate gradient, Jacobi, multigrid |
| {numerical_method} | finite difference, spectral, Crank-Nicolson |
| {factorization_method} | SVD, LU, eigenvalue decomposition |
| {conv_layer_count} | 2, 6, 12 |
| {neuron_count} | 64, 512, 2048 |
| {sparse_format} | CSR, ELL, HYB |
| {nbody_algorithm} | Barnes-Hut, brute force, particle mesh |
| {filter_type} | Gaussian, Sobel, Gabor |
| {filter_size} | 3, 7, 15 |
| {resolution} | 720p, 1080p, 4K |
| {segmentation_algorithm} | watershed, region growing, U-Net |
| {signal_transform} | FFT, wavelet, Hilbert |
| {optimization_algorithm} | Adam, simulated annealing, particle swarm |
| {crypto_algorithm} | AES, RSA, Argon2 |
| {cracking_method} | brute force, dictionary attack, rainbow table |
| {hash_algorithm} | SHA-256, BLAKE3, Bcrypt |
| {data_structure} | binary tree, hash table, bloom filter |
| {collision_strategy} | linear probing, cuckoo hashing, separate chaining |

### A.8.1 PROMPT TEMPLATES FOR SYNTHETIC CUDA CODE GENERATION

The prompts used, listed below, were designed with variations in computation patterns (e.g., memory operations, thread sync, and control flow) and domains (e.g., ML, simulation, graphics), with the intent of diversifying the scope of the synthetic samples.

#### BASIC OPERATIONS

```
1. Implement a CUDA kernel for {size}D FFT (Fast Fourier Transform).
   Optimize for {optimization}.
2. Generate a CUDA implementation for {size}D stencil computation with
   radius {radius}. Optimize for {optimization}.
3. Write a CUDA kernel for parallel reduction to compute the {operation}
   of an array of size {size}. Focus on {optimization}.
4. Create a CUDA implementation for convolution operation with a {size}x{
   size} filter. Focus on {optimization} optimization.
5. Generate a CUDA kernel for matrix multiplication of two matrices A and
    B of size {size}x{size}. Include error handling and optimize for {
   optimization}.
```

#### GRAPH ALGORITHMS

```
1. Write a CUDA implementation for graph coloring of a graph with {size}
   nodes. Focus on {optimization}.
2. Implement a CUDA kernel for community detection in a graph with {size}
    nodes using the {community_algorithm} algorithm.
3. Implement a CUDA kernel for graph processing that computes {algorithm}
    on a graph with {size} nodes. Optimize for {optimization}.
4. Generate a CUDA kernel for finding strongly connected components in a
   directed graph with {size} nodes. Optimize for {optimization}.
5. Create a CUDA implementation for breadth-first traversal on a graph
   with {size} nodes stored in {graph_format}. Optimize for {
   optimization}.
```

#### SCIENTIFIC COMPUTING

```
1. Write a CUDA implementation for {size}D fluid simulation using {method
   }. Focus on {optimization}.
2. Create a CUDA kernel for Monte Carlo simulation of {size} paths for
   option pricing. Focus on {optimization}.
3. Implement a CUDA solver for {size}x{size} sparse linear system using {
   linear_solver}. Focus on {optimization}.
4. Generate a CUDA implementation for {size}D heat equation solver using
   {numerical_method}. Optimize for {optimization}.
5. Create a CUDA kernel for molecular dynamics simulation of {size}
   particles using {md_algorithm}. Optimize for {optimization}.
```

#### MACHINE LEARNING

```
1. Generate a CUDA kernel for k-means clustering of {size} data points in
    {dimension}D space. Optimize for {optimization}.
2. Implement a CUDA kernel for {size}x{size} matrix factorization using {
   factorization_method}. Optimize for {optimization}.
3. Create a CUDA implementation for computing attention mechanism in a
   transformer with {size} tokens. Focus on {optimization}.
4. Implement a CUDA kernel for backpropagation in a convolutional neural
   network with {conv_layer_count} conv layers. Optimize for {
   optimization}.
5. Write a CUDA implementation for training a neural network with {
   layer_count} layers and {neuron_count} neurons per layer. Focus on {
   optimization}.
```

SPARSE OPERATIONS

```
1. Generate a CUDA kernel for sparse FFT computation. Optimize for {
   optimization}.
2. Implement a CUDA kernel for sparse tensor operations with {size} non-
   zero elements. Optimize for {optimization}.
3. Write a CUDA implementation for sparse convolution with {size}x{size}
   filter on sparse input. Focus on {optimization}.
4. Create a CUDA implementation for sparse matrix-matrix multiplication
   in {sparse_format} format. Focus on {optimization}.
5. Generate a CUDA kernel for sparse matrix-vector multiplication where
   the matrix has approximately {size} non-zero elements. Optimize for {
   optimization}.
```

SIMULATION

```
1. Generate a CUDA kernel for cloth simulation with {size}x{size} grid.
   Optimize for {optimization}.
2. Write a CUDA implementation for raytracing of a scene with {size}
   objects. Focus on {optimization}.
3. Create a CUDA implementation for {algorithm} of {size} particles in a
   {dimension}D space. Focus on {optimization}.
4. Create a CUDA implementation for fluid-structure interaction with {
   size} boundary elements. Focus on {optimization}.
5. Implement a CUDA kernel for N-body simulation of {size} particles
   using {nbody_algorithm}. Optimize for {optimization}.
```

IMAGE AND SIGNAL PROCESSING

```
1. Create a CUDA implementation for feature extraction from {size}x{size}
   images. Focus on {optimization}.
2. Generate a CUDA kernel for image segmentation using {
   segmentation_algorithm}. Optimize for {optimization}.
3. Write a CUDA implementation for real-time video processing of {
   resolution} frames. Focus on {optimization}.
4. Implement a CUDA kernel for signal processing with {size}-point {
   signal_transform}. Optimize for {optimization}.
5. Implement a CUDA kernel for image filtering using {filter_type} filter
   of size {filter_size}x{filter_size}. Optimize for {optimization}.
```

OPTIMIZATION ALGORITHMS

```
1. Implement a CUDA kernel for simulated annealing with {size} states.
   Optimize for {optimization}.
2. Generate a CUDA kernel for genetic algorithm with population size {
   size}. Optimize for {optimization}.
3. Write a CUDA implementation for {optimization_algorithm} with {size}
   variables. Focus on {optimization}.
4. Write a CUDA implementation for gradient descent optimization with {
   size} parameters. Focus on {optimization}.
5. Create a CUDA implementation for particle swarm optimization with {
   size} particles in {dimension}D space. Focus on {optimization}.
```

### CRYPTOGRAPHY AND SECURITY

```
1. Generate a CUDA kernel for homomorphic encryption operations. Optimize
     for {optimization}.
2. Write a CUDA implementation for secure hashing using {hash_algorithm}.
     Focus on {optimization}.
3. Generate a CUDA kernel for {crypto_algorithm} encryption/decryption.
    Optimize for {optimization}.
4. Create a CUDA implementation for blockchain mining with difficulty {
    size}. Focus on {optimization}.
5. Implement a CUDA kernel for password cracking using {cracking_method}.
       Optimize for {optimization}.
```

### DATA STRUCTURES

```
1. Create a CUDA implementation for priority queue with {size} elements.
    Focus on {optimization}.
2. Create a CUDA implementation for {data_structure} with {size} elements.
     Focus on {optimization}.
3. Implement a CUDA kernel for operations on a B-tree with {size} nodes.
    Optimize for {optimization}.
4. Generate a CUDA kernel for skip list operations with {size} elements.
    Optimize for {optimization}.
5. Write a CUDA implementation for hash table with {size} buckets using {
    collision_strategy}. Focus on {optimization}.
```

### A.8.2 QUALITATIVE COMPARISON WITH OTHER LLMS

We highlight several cases where CASS-7B outperforms existing LLMs such as Claude, Qwen-Coder, and GPT-4o in faithfully transpiling CUDA to HIP. For example, in one instance, CASS-7B correctly transpiled the CUDA code while preserving the exact string constants from the original program, including the label CUDA in the output format string. Maintaining these strings is essential for preserving the intended user-facing behavior, particularly in logging or debugging scenarios where clarity and consistency matter. In contrast, Claude, Qwen-Coder, and GPT4o unnecessarily altered the string to say HIP, despite the output still originating from a CUDA kernel. This substitution introduces a semantic error, as the original string refers to CUDA, not HIP, and should remain unchanged.

**CASS-7B**

```
printf("tanh(%f) = %f CUDA vs %f (CPU)\n",
       h_input[idx], h_output[idx], tanh(h_input[idx]));
```

**Claude, Qwen-Coder, GPT4o**

```
printf("tanh(%f) = %f (HIP) vs %f (CPU)\n",
       h_input[idx], h_output[idx], tanh(h_input[idx]));
```

In another example, CASS-7B retained the classical CUDA-style kernel launch syntax using triple angle brackets (<<<...>>>), while also ensuring that the generated code remained compilable by correctly including the required HIP header <hip/hip_runtime.h>. This demonstrates a high degree of structural fidelity to the source code, which is especially important for developers familiar with standard CUDA conventions. In contrast, other models such as Claude and Qwen-Coder replaced the launch expression with the HIP-specific macro hipLaunchKernelGGL, which, while functionally valid, deviates from the original representation. More critically, they failed to include the necessary HIP header, rendering the output uncompilable. This example highlights how CASS-7B goes beyond syntactic accuracy to produce code that is both faithful to the original structure and immediately usable in a real compilation setting.

**CASS-7B**

```
#include <hip/hip_runtime.h>
#include <iostream>
...
add<<<(N + 255) / 256, 256>>>(d_a, d_b, d_c, N);
```

**Claude, Qwen-Coder**

```
#include <iostream>
...
hipLaunchKernelGGL(add, (N + 255) / 256, 256, 0, 0, d_a, d_b, d_c, N);
```

Lastly, when verifying numerical correctness, CASS-7B preserved the original logging behavior by correctly emitting output to std::cout, as in the source code. This choice maintains consistency with the original program's semantics, especially in distinguishing between standard output and error streams; important in contexts where output may be redirected or parsed. In contrast, GPT-4o unnecessarily altered the output stream to std::cerr, which, while syntactically valid, changes the runtime behavior of the program. Such a change could lead to unexpected side effects in downstream tools or logging pipelines. This example further demonstrates CASS-7B's attention to both structural and behavioral fidelity in its translations.

**CASS-7B**

```
std::cout << "Error at element " << i << ": " << h_output[I]
          << " vs. expected " << h_reference[i] << std::endl;
```

**GPT4o**

```
std::cerr << "Error at element " << i << ": " << h_output[i]
          << " vs expected " << h_reference[i] << std::endl;
```

