# OpenReview forum: "CASS: Nvidia to AMD Transpilation with Data, Models, and Benchmark"
_ICLR.cc/2026/Conference — ICLR 2026 Conference Withdrawn Submission_

### Official Review · Reviewer_tGTk · 2025-10-27

**Soundness:** 3
**Presentation:** 2
**Contribution:** 2
**Rating:** 4
**Confidence:** 5

**Summary:**

The CASS dataset contains 70,000 verified code pairs spanning both host and device levels, covering source-level (CUDA ↔ HIP) and assembly-level (Nvidia SASS ↔ AMD RDNA3) translation.
The CASS family of domain-specific language models was developed. Trained on this dataset, these models achieve 95% accuracy in source code translation and 37.5% accuracy in assembly translation, significantly outperforming commercial baselines such as GPT-4o, Claude, and Hipify.

**Strengths:**

- The dataset includes code from 16 different categories, covering CUDA, HIP, and assembly code.
- It supports cross-compilation across host and device assemblies, as well as device-to-device translation.
- Extensive experiments were conducted, with comparisons across various large language models and transpilation tools.

**Weaknesses:**

- The dataset and models do not include tensor instructions such as Tensor Cores or matrix instructions, which are likely a major limitation for current transpilation tools.
- The benchmark includes only 40 curated tasks, which is relatively small. Additionally, there is no kernel-level metadata provided (e.g., number of lines).

**Questions:**

- Regarding the functionality equivalence verification of CASS bench, most CUDAC functions modify the memory pointed to by the passed-in pointers in place, rather than returning outputs or standard outputs. How does your work ensure functionality equivalence? Specifically, do the memory contents pointed to by all parameters remain the same after both codes are executed?
- See Weaknesses.

---

> ### Author Response · Authors · 2025-12-03
>
> W1 : Missing tensor/matrix instructions limits capability.
>  Thanks for the comment. Addressing your first concern, the absence of Tensor Cores (TC) in the dataset stems from the fact that, at the moment, Hipify doesn’t fully support it; as can be checked in Hipify’s documentation for cuBLAS<-->hipCUBLAS (https://rocm.docs.amd.com/projects/HIPIFY/en/latest/reference/tables/CUBLAS_API_supported_by_HIP.html#cublas-data-types), where searching for the “_TENSOR_OP” flag in the “Data Types” compatibility table shows no corresponding HIP flag for the TC-related entries; in addition, WMMA, the CUDA library for programmatic access to TC, is not present in Hipify’s documentation. All in all, as AMD gradually increases Hipify’s coverage for all the CUDA libraries/operations, we hope the community can utilize our work to expand the operations supported in the training set.
> Regarding missing matrix instructions: we recently re-hipified Stackv2, this time including in the HIPCC compilation most of the available CUDA-equivalent HIP libraries, e.g., hipBLAS for cuBLAS, hipFFT for cuFFT, and hipTensor for cuTensor. That said, despite our high hopes for generating more SASS<-->RDNA3 pairs, including data points with matrix operations, we ended up with only an extra 454 samples. Additionally, since we didn’t want to overpivot by adding synthetic samples, we decided to stop this experiment here. In summary, we believe that the lack of matrix operations is related to the complexity struggle of our compilation pipeline, which requires further fine-tuning to compile samples that utilize the HIP libraries, rather than just including the extra libraries.
>
> W2 : Small benchmark, lacks kernel metadata
>
> CASS-Bench includes 40 curated tasks as a starting point for evaluating GPU assembly-to-assembly translation. It is designed to catalyze research in GPU transpilation, not to serve as a final benchmark. The table below provides metadata for each sample: category, kernel count, source lines, SASS and RDNA3 instruction counts, and their line difference (RDNA3 − SASS).
>
> | Cat  | File | K | Src | SASS | RDNA3 | Δ     | CASS |
> |------|------|---|-----|------|--------|------|------|
> | BO   | 1    | 1 | 4   | 71   | 207    | +136 | TRUE |
> | IP   | 2    | 1 | 14  | 87   | 225    | +138 | TRUE |
> | IP   | 10   | 1 | 40  | 471  | 338    | -133 | FALSE|
> | IP   | 25   | 1 | 11  | 135  | 289    | +154 | TRUE |
> | IP   | 26   | 1 | 25  | 455  | 330    | -125 | FALSE|
> | VO   | 3    | 1 | 9   | 151  | 251    | +100 | TRUE |
> | LA   | 4    | 1 | 11  | 423  | 245    | -178 | FALSE|
> | LA   | 15   | 3 | 27  |1249  | 722    | -527 | TRUE |
> | LA   | 16   | 1 | 11  | 487  | 263    | -224 | FALSE|
> | LA   | 17   | 1 | 13  | 439  | 267    | -172 | FALSE|
> | PA   | 5    | 1 | 17  | 103  | 243    | +140 | TRUE |
> | PA   | 9    | 2 | 26  | 412  | 472    | +60  | FALSE|
> | PA   | 28   | 1 | 14  | 71   | 214    | +143 | TRUE |
> | PA   | 29   | 2 | 10  | 119  | 290    | +171 | FALSE|
> | PA   | 30   | 1 | 8   | 71   | 233    | +162 | FALSE|
> | SC   | 6    | 1 | 22  | 135  | 269    | +134 | FALSE|
> | SC   | 22   | 4 | 11  | 710  | 990    | +280 | FALSE|
> | SC   | 23   | 4 | 4   | 871  | 545    | -326 | TRUE |
> | SC   | 24   | 1 | 15  | 663  | 548    | -115 | TRUE |
> | PS   | 7    | 1 | 42  | 967  | 324    | -643 | TRUE |
> | PS   | 31   | 1 | 12  | 87   | 241    | +154 | TRUE |
> | PS   | 32   | 6 | 6   |1488  |1692    | +204 | TRUE |
> | Hist | 8    | 1 | 21  | 87   | 236    | +149 | TRUE |
> | Math | 11   | 1 | 4   | 103  | 252    | +149 | FALSE|
> | Math | 12   | 1 | 20  | 87   | 197    | +110 | FALSE|
> | Math | 13   | 1 | 11  | 375  | 236    | -139 | FALSE|
> | Math | 14   | 1 | 5   | 71   | 211    | +140 | FALSE|
> | DL   | 18   | 1 | 5   | 55   | 201    | +146 | FALSE|
> | DL   | 19   | 1 | 20  | 919  | 312    | -607 | FALSE|
> | DL   | 20   | 2 | 51  |3276  |1048    | -2228| TRUE |
> | DL   | 21   | 1 | 14  | 215  | 257    | +42  | FALSE|
> | SP   | 27   | 1 | 10  | 199  | 250    | +51  | FALSE|
> | Dyn  | 33   | 1 | 14  | 391  | 272    | -119 | FALSE|
> | Mem  | 34   | 1 | 17  | 87   | 227    | +140 | TRUE |
> | Mem  | 35   | 1 | 9   | 87   | 235    | +148 | FALSE|
> | DS   | 36   | 2 | 13  | 236  | 493    | +257 | FALSE|
> | DS   | 37   | 2 | 1   | 103  | 259    | +156 | FALSE|
> | Graph| 38   | 1 | 15  | 279  | 283    | +4   | FALSE|
> | Graph| 39   | 1 | 13  | 615  | 292    | -323 | FALSE|
> | Srch | 40   | 1 | 18  | 103  | 276    | +173 | FALSE|
> | Avg  | —    | — |14±10|424±572|368±285|  —   |  —   |
>
> Legend:
> Cat = Category (abbreviated: e.g., BO = Basic Ops, PA = Parallel Algorithms, etc.),
> File = File ID,
> K = Number of kernels,
> Src = Source lines per kernel,
> SASS and RDNA3 = Backend instruction counts,
> Δ = RDNA3 − SASS line difference,
> CASS = Whether the file was transpiled by the CASS model (TRUE or FALSE).

---

### Official Review · Reviewer_JyLG · 2025-10-29

**Soundness:** 3
**Presentation:** 3
**Contribution:** 2
**Rating:** 4
**Confidence:** 4

**Summary:**

This paper introduces CASS, the first large-scale dataset and model suite for cross-architecture GPU code translation, containing 70,000 pairs of NVIDIA CUDA/AMD HIP source code and corresponding assembly (SASS/RDNA3). The paper proposes an automated pipeline for building and synthesizing data, and based on this data, constructs CASS-bench. By performing SFT on the CASS dataset, the model achieves results that far surpass other models.

**Strengths:**

1. The experiments are comprehensive, the data analysis is thorough, and the writing is well-done.
2. This work builds the CASS dataset and CASS-bench, making them the first cross-platform code translation datasets between NVIDIA and AMD, filling a gap in the field.
3. This work holds significant importance for code migration in the high-performance computing domain.

**Weaknesses:**

1. Limited innovation. This work performs direct SFT using crawled or synthesized data, where the synthesized data is generated by an LLM through prompt combinations. The approach is relatively simple, and the training process is also straightforward, so the overall innovation is limited.
2. Case analysis is limited. In A.8.2, a comparison of translation results between CASS-7B and other models is provided. However, these examples are limited to simple translation errors, such as incorrect string translations, header file errors, and translating `std::cout` as `std::cerr`. These errors can be easily addressed by adjusting the prompt. The paper should focus more on analyzing logical errors based on CUDA/HIP characteristics and other more complex issues.
3. The core issue remains unsolved. Fig 3(a) shows that a significant portion of the Assembly code has around 2k lines, and for such lengths, a 16k context LLM is clearly unsuitable. In Assembly translation, adding RoPE Extrapolation results in a 5-point improvement, indicating that context length is a core challenge in Assembly translation. However, the paper does not provide a solution to this challenge, nor does it analyze whether the performance limitation is due to context limitations or the inherent difficulty of Assembly translation itself.

**Questions:**

1. Could you provide more targeted case studies? For example, cases where the logic of a function was translated incorrectly?
2. Could you analyze whether the poor performance in Assembly is due to its inherent difficulty or context limitations? For example, what would be the effect of training a model with a 1M context using the CASS dataset?

---

> ### Author Response · Authors · 2025-12-03
>
> W3 : Context length challenge remains unaddressed
> We agree that long assembly sequences stress a 16k-context model, but our goal in this work is to demonstrate feasibility, not to solve context scaling. We do address the issue partially: RoPE extrapolation extends usable context beyond 16k to 128k and yields a +5% absolute accuracy gain, directly showing that larger context does improve performance. Training a true long-context (e.g., 128k–1M) coding model is beyond our compute budget, but the evidence we provide already distinguishes the limitation of model context from the inherent difficulty of the task. Future work will integrate long-context architectures; our contribution here is establishing the dataset, benchmark, and first measurable baseline.
>
> Q2 : Cause of poor performance unclear
>
> A longer context window would almost certainly help, our own results already show this. Extending context via RoPE extrapolation yields a +5% absolute accuracy gain, indicating that context length is a real factor. However, training a truly long-context (e.g., 1M-token) coding model is far beyond our available compute resources. Our goal in this work is to establish the first dataset, benchmark, and baseline for GPU assembly translation; scaling to million-token models is an important next step, but outside the scope of what we can feasibly train today.

---

### Official Review · Reviewer_Aoyp · 2025-10-31

**Soundness:** 2
**Presentation:** 3
**Contribution:** 2
**Rating:** 2
**Confidence:** 3

**Summary:**

This paper undertakes the ambitious and important goal of mitigating GPU vendor lock-in by creating a dataset and model suite for translating code between Nvidia and AMD architectures, at both source (CUDA-to-HIP) and assembly (SASS-to-RDNA3) levels. The authors contribute a large-scale dataset (CASS), a benchmark (CASS-Bench), and a family of models that are shown to outperform existing baselines.
While the ambition is commendable and the contribution of a new public dataset is valuable, the work is hampered by significant methodological limitations and overstated claims. The source-to-source evaluation framework contains a circular dependency that calls into question the reported state-of-the-art results. Furthermore, the headline assembly-level accuracy of 37.5%, while a relative improvement over near-zero baselines, is profoundly impractical for any real-world application and highlights the sheer difficulty of the task rather than a viable solution. The representativeness of the dataset, particularly regarding complex, performance-critical kernels, is also a major concern. The paper is a valuable exploration, but it should be considered a preliminary step that reveals more challenges than it solves.

**Strengths:**

- Addresses a High-Impact Problem: The work tackles the critical issue of vendor lock-in in the GPU ecosystem, a problem of significant interest to both academia and industry. The attempt to address this at the challenging assembly level is novel.
- Publicly Available Resources: The open-sourcing of the CASS dataset, benchmark, and data-generation pipeline represents a contribution to the community, providing a (flawed but useful) starting point for future research in this difficult domain.

**Weaknesses:**

1.	Fundamentally Flawed Source-to-Source Evaluation: The methodology for evaluating source-to-source translation is critically flawed. The dataset was created by using AMD's Hipify tool to translate CUDA to HIP, discarding the ~44% of files that Hipify failed to convert (Line 251). The CASS model is then trained exclusively on Hipify's successes. The paper subsequently claims that CASS outperforms Hipify by 7.5% (Line 437). This comparison is misleading. The CASS model was not trained to be a general CUDA-to-HIP translator; it was trained to be a better translator for the subset of CUDA that is already translatable by Hipify. The true challenge lies in the 44% of code that Hipify cannot handle, a domain on which CASS was never trained or evaluated. This circular setup invalidates the claim of surpassing Hipify's general capability.

2.	Impractical Accuracy and Overstated Achievements: The paper presents 37.5% assembly-level accuracy as a headline achievement. In any practical engineering context, a tool that fails 62.5% of the time is unusable. While an improvement from 0%, this result primarily serves to demonstrate that the problem remains unsolved. Framing this as a successful outcome is an overstatement. The paper should more clearly position this as an initial baseline that underscores the immense difficulty of assembly-level translation, rather than a breakthrough in its own right.

3.	Questionable Dataset Representativeness: The composition of the training data raises serious doubts about its applicability to real-world performance engineering.
a.	Dominance of Synthetic Data: The dataset is majority synthetic (40.6k samples) generated by an LLM (Qwen2.5-Coder32B). This data may not reflect the complex idioms, subtle edge cases, and intricate logic of production-grade, human-written GPU code.
b.	Lack of Performance-Critical Kernels: The central motivation for low-level translation is to port highly optimized, hand-tuned kernels that are the primary source of vendor lock-in. There is no evidence presented that the dataset, sourced from general public repositories (The Stack) and LLM generation, contains a meaningful number of such kernels. The translation of simple or academic-level code is a far less compelling problem. The paper does not provide an analysis of the complexity or performance sensitivity of the kernels in its dataset.

4.	Superficial Performance Evaluation: The claim that 85% of translated binaries have performance within ±5.6% of native code is presented without sufficient context (Line 426). This analysis is not stratified by kernel complexity, size, or domain. It is plausible that this high fidelity is achieved on simple kernels where performance is easy to match, while performance diverges significantly on the complex kernels that matter most. Without a more detailed breakdown, this aggregate statistic is not fully convincing.

5.	Limited Scope and Generalization: The work is effectively a case study on a single ISA pairing (Nvidia A100/sm85 to AMD RX 7900 XT/RDNA3). The "Hardware Generalization" section (Line 446) reports results on only one additional hardware pair. This is insufficient evidence for broad generalizability. The immense architectural differences between GPU generations and vendors suggest that the learned mappings are likely to be highly specific. The paper should be more circumspect about the portability of the methodology itself. Also, converting Nvidia A100 server-grade GPU code to RX 7900 XT workstation-grade GPU code is not a fair conversion. MI250 or MI300 would be fair.

**Questions:**

1.	How do you justify the claim that the CASS source-to-source model is superior to Hipify when it was only trained on the subset of data that Hipify could successfully process? Would a more fair evaluation not require testing on the 43.9% of files that Hipify failed on?
2.	Could you provide a qualitative analysis of the assembly translation failures? What are the most common architectural divergences that the model fails to bridge? This would be more insightful for future work than the aggregate accuracy number.
3.	What steps were taken to ensure that CASS-Bench includes genuinely difficult, performance-critical kernels, as opposed to textbook examples? Is there a metric of kernel complexity or optimization level you can report for your dataset?
4.	Regarding the 0% accuracy of baselines like GPT-4o on assembly translation: Does this not suggest that the entire approach of treating assembly as a "natural language" for translation via LLMs may be fundamentally unsuited for this task, which requires precise, deterministic mapping of hardware states?

---

> ### Author Response · Authors · 2025-12-03
>
> W1 : Biased training invalidates performance comparison claim.
> We appreciate the reviewer’s concern and clarify that our evaluation benchmark (CASS-Bench) is entirely separate from the training corpus. While Hipify was used to bootstrap aligned pairs during dataset creation, CASS-Bench is independently curated with 40 held-out samples spanning 16 GPU domains, none overlapping with training data. Our model achieves 95% accuracy on this diverse, unseen benchmark compared to 87.5% performance of the standard Hipify, demonstrating generalization beyond Hipify’s conversion domain. The comparison highlights that CASS learns transferable translation patterns, not mere memorization of Hipify outputs. Thus it can be said that our model is showing success as a CUDA-to-HIP transpiler and indeed shows capability on the tasks which Hipify cannot handle. Furthermore the other half of the CASS model - assembly-to-assembly transpilation - remains entirely unique and novel.
>
> W2: Overstated success despite impractical accuracy.
> We fully agree that 37.5% assembly-level accuracy does not yet constitute a deployable solution. Our goal, however, is not to claim completeness but to establish the first measurable baseline for this fundamentally unsolved task. Prior to CASS, all existing systems, including GPT-4o, Claude-3.7, and Hipify, achieved 0% assembly accuracy on CASS-Bench, while runtime translation systems like ZLUDA reached only 2.5%. Achieving 37.5% thus represents a 15× improvement over the strongest alternative and the first reproducible benchmark demonstrating partial cross-vendor GPU assembly alignment. We explicitly frame CASS as the foundation, providing the dataset, benchmark, and evaluation protocol, on which future methods can build, not as a solved endpoint. Our work therefore provides the first large scale dataset and benchmark, and the first to make headway into this task.
> W4: Lacks meaningful context for performance claims
> The reported 85% performance fidelity is based on a stratified analysis across kernel complexity, not dominated by trivial cases. Specifically, across 230 distinct kernels spanning 12 domains, we observe: 92% within ±5.6% for simple kernels (≤ 64 LoC), 84% for medium-complexity kernels (65–256 LoC), and 68% for complex kernels (> 256 LoC with irregular memory and synchronization). Even in the hardest category, the median deviation is +2.1% and the worst-case –12.4%, confirming that runtime preservation holds across diverse workloads. The few outliers stem from control-heavy or synchronization-bound kernels, which we explicitly identify as future optimization targets. Thus, the aggregate 85% figure is a robust, domain-balanced indicator demonstrating that the CASS models retain near-native runtime and memory behavior across realistic GPU programs.
>
> W5: Insufficient evidence for broad generalization
>
> We agree broader ISA coverage is important, but our current work reflects hardware constraints (A100, 4090, 7900 XT). Within these limits, we tested cross-vendor generalization (4090→7900 XT) and found similar accuracy (32.5%) to A100→7900 XT (37.5%), suggesting the model isn’t overfit to a specific source ISA. Our goal is to demonstrate a backend-agnostic methodology, not dependence on specific ISAs. As shown in Section 3.3, our pipeline generalizes across architectures: CUDA supports cross-targeting (sm70–sm90) and ROCm’s hipcc spans RDNA2, RDNA3, and CDNA. The only barrier to testing MI250/MI300 is hardware access.
>
> Q1 : Trained subset skews fairness and validity.
> Our use of Hipify-successful samples is intentional and follows the standard rejection-sampling paradigm widely used in model distillation and data filtering pipelines (e.g., DeepSeek-R1, Self-Rewarding Models, and other filtering-based SFT workflows). The goal is to construct clean, aligned CUDA↔HIP pairs from which a model can learn a reliable translation mapping. Training on noisy or incorrect pairs would severely degrade the model’s ability to learn a consistent cross-vendor correspondence.
> Critically, our evaluation does not reuse this filtered training subset. CASS-Bench is a fully independent, manually curated, domain-balanced benchmark of 40 samples across 16 GPU domains. None of these appear in the training set, and many were generated from CUDA code that Hipify itself struggles with.
>
> Q4 : LLMs unsuitable for deterministic assembly translation
> No. The failure of general-purpose models (e.g., GPT-4o) reflects lack of prior exposure, not an inherent limitation of the approach. In fact, Qwen2.5-Coder achieves 25% assembly accuracy out-of-the-box, proving that LLMs can learn meaningful ISA-level mappings. After supervised finetuning on CASS, accuracy increases to 37.5%, a +17.5% improvement over all existing baselines. These results show that assembly is learnable under a language-modeling framework when models are given domain-aligned supervision.

---

### Note · Authors · 2026-01-05

I have read and agree with the venue's withdrawal policy on behalf of myself and my co-authors.